# Antenatal care service utilization disparities between urban and rural communities in Ethiopia: A negative binomial Poisson regression of 2019 Ethiopian Demography Health Survey

**Fitsum Endale**[1]*, **Belay Negassa**[2], **Tizita Teshome**[3], **Addisu Shewaye**[4], **Beyadiglign Mengesha**[5], **Endale Liben**[6], **Senahara Korsa Wake**[7]

1 School of Public Health, College of Medicine and Health Sciences, Wachemo University, Hossana, Ethiopia, 2 Department of Environmental Health, College of Health Sciences and Medicine, Dilla University, Dilla, Ethiopia, 3 Department of Environmental Health Science and Technology, Institute of Health, Jimma University, Jimma, Ethiopia, 4 Yem Saja Secondary High School, Saja, Ethiopia, 5 Jimma City Municipality Office, Jimma, Ethiopia, 6 Ethiopian Public Health Institute, Addis Ababa, Ethiopia, 7 College of Natural and Computational Science, Ambo University, Ambo, Ethiopia

* fitsumale@gmail.com

## Abstract

### Background

Although there have been consistent improvements in maternal mortality, it remains high in developing countries due to unequal access to healthcare services during pregnancy and childbirth. Thus, this study aimed to further analyze the variations in the number of antenatal care utilizations and associated factors among pregnant women in urban and rural Ethiopia.

### Methods

A total of 3962 pregnant women were included in the analysis of 2019 Ethiopian Demographic and Health Survey data. A negative binomial Poisson regression statistical model was used to analyze the data using STATA version 14.0. An incident rate ratio with a 95% confidence interval was used to show the significantly associated variables.

### Results

Of the 3962 (weighted 3916.67) pregnant women, about 155 (15.21%) lived in urban and 848 (29.29%) rural residences and did not use antenatal care services in 2019. Women age group 20–24 (IRR = 1.30, 95%CI:1.05–1.61), 25–29 (IRR = 1.56, 95%CI:1.27–1.92), 30–34 (IRR = 1.65, 95%CI:1.33–2.05), and 35–39 years old (IRR = 1.55, 95%CI:1.18–2.03), attending primary, secondary, and higher education (IRR = 1.18, 95%CI:1.07–1.30), (IRR = 1.26, 95%CI:1.13–1.42) and (IRR = 1.25, 95%CI:1.11–1.41) respectively, reside in middle household wealth (IRR = 1.31, 95%CI:1.13–1.52), richer (IRR = 1.45, 95%CI:1.26–1.66)

**Data Availability Statement:** The current study used women with completed interviews called the Individual Recode (IR) survey dataset, which originated from the official Demographic and Health Survey (DHS) program database. The DHS program upholds stringent restrictions against data sharing with unapproved parties in order to protect the privacy and confidentiality of survey participants. As a result, in order to gain access to the data, researchers must follow the program's data protection guidelines and pass a rigorous review procedure. Requests for data can be sent to www.dhsprogram.com by interested parties. However, upon reasonable request and with the DHS's approval, the datasets developed and analyzed during the current investigation may be made available. The data may only be made public with the explicit consent of Jimma University's institutional review board (IRB) research committee in accordance with the regulations of the DHS program. The committee can be reached at henok.gulilat@ju.edu.et by interested parties.

**Funding:** The author(s) received no specific funding for this work.

**Competing interests:** The authors have declared that no competing interests exist.

and richest (IRR = 1.68, 95%CI:1.46–1.93) increases the number of antenatal care utilization among urban residences.

While attending primary (IRR = 1.34, 95%CI:1.24–1.45), secondary (IRR = 1.54, 95% CI:1.34–1.76) and higher education (IRR = 1.58, 95%CI:1.28–1.95), following Protestant (IRR = 0.76, 95%CI:0.69–0.83), Muslim (IRR = 0.79, 95%CI:0.73–0.85) and Others (IRR = 0.56, 95%CI:0.43–0.71) religions, reside in poorer, middle, richer, and richest household wealth (IRR = 1.51, 95%CI:1.37–1.67), (IRR = 1.66, 95%CI:1.50–1.83), (IRR = 1.71, 95% CI:1.55–1.91) and (IRR = 1.89, 95%CI:1.72–2.09) respectively, being married and widowed/separated (IRR = 1.85, 95%CI:1.19–2.86), and (IRR = 1.95, 95%CI:1.24–3.07) respectively were significantly associated with the number of antenatal care utilization among rural residences.

## Conclusion

The utilization of antenatal care is low among rural residents than among urban residents. To increase the frequency of antenatal care utilization, health extension workers and supporting actors should give special attention to pregnant women with low socioeconomic and educational levels through a safety-net lens.

## Introduction

Maternal and child health is an extended global challenge that has been considered in the United Nations' Sustainable Development Goals (SDGs) [1]. The world has made steady progress in reducing maternal mortality; according to UN inter-agency estimates from 2000 to 2020, the global maternal mortality ratio (MMR) declined by 34% from 339 deaths to 223 deaths per 100,000 live births. However, it remains high in developing countries due to a lack of access to healthcare during pregnancy and childbirth [2]. Utilization of healthcare services is a key consequential predictor of infant and maternal outcomes [3]. Antenatal care (ANC) is still an important healthcare tool for reducing the risk of stillbirths, preterm labor, and pregnancy complications because it serves as a platform for key healthcare tasks such as health promotion, screening, and diagnosis, as well as disease prevention [3, 4].

ANC refers to the medical procedures and care provided during pregnancy [5], including the clinical assessment of the pregnant woman and her fetus, aimed at achieving a favorable outcome for both the mother and child [6]. The World Health Organization (WHO) recommended at least four ANC visits for normal pregnant women and more than four visits for women with complications. The recommended visits are in the first trimester: the first visit for counseling and screening for risk factors as well as medical conditions; in the second and third trimesters: two visits to monitor maternal and fetal conditions; and one additional visit in case it is an elongated pregnancy [4]. During the ANC visits, pregnant women undergo screening for pre-existing health conditions, receive diagnoses, and are provided with suitable interventions. The women and their families also receive behavioural change communication focusing on personal hygiene, nutrition, and the utilization of available services and interventions [4, 7, 8]. Even though ANC utilization has increased overtime, it is still low compared to WHO's guideline, which also indicate that the use of ANC varies with a huge underutilization among pregnant women in low and middle-income countries [9, 10].

In Ethiopia, ANC services are available in both urban and rural areas [11], and pregnant women are encouraged to make at least four antenatal care visits until delivery, one within each trimester [12]. Due to healthcare system reforms, particularly maternal health policy, Ethiopian pregnant women currently have more options to visit for ANC utilization since ANC services are available at public and private healthcare facilities. Even though ANC services are available at every healthcare facility, there is a significant gap between regions and social groups within one region in the utilization of ANC [13].

Similarly, studies have reported a higher ANC utilization among urban women than rural women in Ethiopia. For instance, in all surveys from 2000 until 2016, Ethiopian Demographic and Health Surveys (EDHS) further analyses showed that women from urban areas had more ANC visits than women from rural areas (67.2% vs. 21.9% in 2000; 69.7% versus 24.3% in 2005; 76.5% vs. 36.8% in 2011; and 90% vs. 58% in 2016) [11, 13]. Simultaneously, studies have been conducted to identify determinant factors associated with ANC utilization in Ethiopia, and it is evidenced that ANC utilization varies based on mothers' educational status, age, exposure to media, occupational type, wealth index, residential place, family size, ease of access to healthcare facilities, and accessibility of ANC services [14–27].

Despite the inequalities in ANC service utilization among urban and rural women [13], there is limited evidence that shows the current disparities in ANC service utilization among this population group using national-level representative data that considers the count model after rural health extension programs have excelled in urban settings. Therefore, the aim of this further DHS data analysis is to compare the level as well as the factors of antenatal care service utilization in rural and urban Ethiopia. The findings of this study could lead to policy recommendations in order to improve maternal healthcare services in general.

## Methods and materials

### Study setting, data source, and period

The study was conducted in Ethiopia, located in north-eastern Africa. The study was based on the intermediate EDHS 2019 dataset, which was conducted by the Central Statistical Agency in collaboration with the Federal Ministry of Health (FMoH) and the Ethiopian Public Health Institute. The survey was conducted from March 21, 2019 to June 28, 2019, based on a nationally representative sample (please check the 2019 EDHS report for more information) [28].

On February 16, 2023, the requested data was obtained from the Demography Health Survey (DHS) program's official database, www.dhsprogram.com, after providing an abstract and stating the justification of the study's objectives via an online form. A cross-sectional study design using secondary data from the 2019 EDHS was conducted.

### The population of the study

A nationally representative sample of 8,663 households provided 8,855 women of reproductive age (aged 15 to 49) as the source population for this study. The study population was 3,979 women who were in the reproductive age group (15–49 years) and had had pregnancy in the previous five years before the data collection period and were living in Ethiopia. Hence, 3,962 (3,916.7 weighted) women's data were extracted from the 2019 intermediate EDHS datasets. After excluding 17 women who had an unknown number of ANC visits (missing data) (Fig 1).

### Sampling procedure

The intermediate EDHS used a complete list of 149,093 enumeration areas (EAs) created for the upcoming Ethiopian population and housing census as a sampling frame. The frame

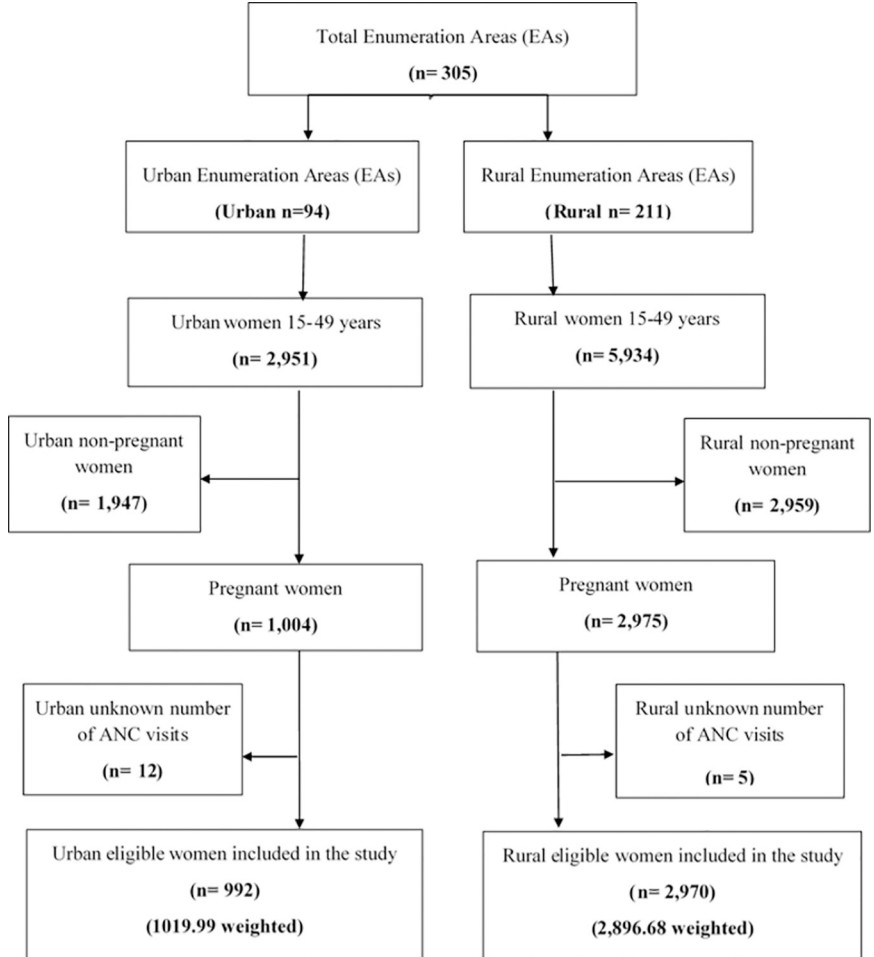

**Fig 1. Study population and sampling procedures to identify the final sample size in the 2019 EMDHS.**

comprises information about the EA type of residence (urban or rural) and the estimated number of residential households.

In light of this, the sample was stratified and selected in two steps. There were 21 sampling strata created in the first step after stratification was done by region and then by urban and rural areas within each region. In each sampling stratum, 305 enumeration areas (EAs) (94 urban and 211 rural) were chosen with a probability proportionate to the EA size. In the second stage, households were selected proportionally from each EA by using a systematic sampling method. Furthermore, the 2019 EMDHS provided details on the survey's design and methodology [28].

## Study variables and measurements

The outcome variable of this study was the number of ANC visits during the last pregnancy between urban and rural women who were in the reproductive age group (15–49 years) and had had pregnancy in the previous five years before the data collection period. The ANC visits were assessed with the "How many times did you receive antenatal care during this pregnancy?" question. The mothers were asked for their number of ANC visits within the last five years of the data collection period as the last pregnancy.

**Table 1. Description of individual and household-level variables.**

| Variables | Description |
|---|---|
| **Individual-level variables** | |
| **Maternal age** | It is the current age of women recoded as 15–19, 20–24, 25–29, 30–34, 35–39, 40–44, and 45–49. |
| **Educational level of the mother** | This is the level of education a woman attained and was recoded as no education, primary, secondary, or higher. |
| **Current marital status** | This is the status of a women, whether she is never married, married, widowed, or separated |
| **Religion** | This variable is the religious group to which the woman associates herself and is recoded as Orthodox, Protestant, Muslim, and other. |
| **Total children ever born** | It is recoded as 0 children, 1–2 children, 3–4 children, and 5 and more. |
| **Wealth index: urban/ rural** | In DHS, the wealth index is calculated using data on a household's ownership of selected assets. Each household asset is assigned a weight score generated through PCA. The resulting asset scores are standardized and summed by household, and individuals are ranked according to the total score of the household in which they reside. Finally, it is grouped as poorest, poorer, middle, richer, and richest. It is recoded as poorest and poorer, middle, richer and richest. |

Factors that were expected to be associated with the number of ANC visits by women in Ethiopia, mainly both individual and household-level factors, were considered. These include; women's age, religion, current marital status, educational level, household wealth index, family size, and number of children (Table 1).

## Data processing and analysis

Data cleaning was conducted to check for consistency with the intermediate EDHS 2019 descriptive report. Recoding, variable generation, labeling and analysis were done using STATA version 14.0. In the EDHS, the sample distribution between urban and rural settings was not proportionate. Therefore, sample weights were used to estimate frequencies to account for disproportionate sampling. The weighting procedure was meticulously explained in the 2019 EDHS report [28].

Frequency, percentage, and mean were calculated for the explanatory and response variables using descriptive statistical analysis. A chi-square test was done to see if there was any association between ANC utilization among urban and rural residences, and a statistically significant difference was observed between the two groups ($\chi 2 = 437.51$, $p < 0.001$), indicating that the factors associated with ANC utilization could be different among rural and urban residences. Therefore, the analysis was conducted separately. The analysis was done for both the urban and rural parts. Finally, incident rate ratio and odds ratio were presented with a 95% CI. Statistical significance was declared at a p-value of less than 0.05.

Since ANC follow-up (dependent variable) is a non-negative integer, most of the recent thinking in the field has used the Poisson regression model as a starting point. The mean and variance should be equal in order to do a Poisson regression. Though the mean and variance in this instance were 2.89 and 5.33, respectively, that is, the data were overly dispersed. So the assumption of Poisson regression is violated. To handle over-dispersion of the data, we have considered the negative binomial Poisson model, the extension of Poisson regression, to have a precise result [29].

The negative binomial model, also known as the Poisson-gamma model, extends the Poisson model to handle potential data over-dispersion. In this model, the assumption is made that the Poisson parameter adheres to a gamma probability distribution. The negative binomial model is derived by rewriting the Poisson parameter for each observation $i$,

where, $\mu_i = exp(\beta X_i + \varepsilon_i)$, and $exp(\varepsilon_i)$ is a gamma-distributed error term with mean 1 and variance K. This term's inclusion permits the variance to differ from the mean in the following ways:

$$Var[y_i] = E[y_i][1 + K[y_i]] + E[y_i] + KE[yi] \tag{1}$$

The following provides the probability mass function for the negative binomial distribution [20]:

$$p(y_i) = \binom{y_i + r - 1}{y_i} p^r (1 - p)^{y_i}, \ r = 0, 1, 2, 3, \ldots \tag{2}$$

The parameter $p$ is the probability of success in each trial, and it is calculated as follows: $p = \frac{r}{\mu_i + r}$ Where, $\mu_i = exp(y)$ is the mean of the observations and $r$ is the inverse of the dispersion parameter (that is, $r = \frac{1}{k}$).

The Poisson regression model emerges as a special case of the negative binomial regression model when the parameter $k$ tends toward zero. The choice between these models hinges on the value of $k$, often termed the over-dispersion parameter. While the negative binomial model effectively addresses over-dispersion, it may encounter limitations in handling situations with an abundance of zero counts.

The comparison of alternative models relies on the maximum likelihood method [30]. In this evaluation, we utilize both Akaike's Information Criterion (AIC) and Bayesian Information Criterion (BIC) criteria to appraise various model specifications. Additionally, the log likelihood ratio test is employed for a similar purpose. These information criteria aim to systematically assess and identify the most suitable model specification derived from the available data. In this study, we focused on the application of classical Poisson regression models only.

## Ethical approval

Since secondary data from archives of demographic and health surveys was used, contacting the DHS Program team for further ethical approval was not necessary. However, after reviewing the abstract of our study proposal submitted through the www.dhsprogram.com portal, the DHS data permission was obtained via email. The data were not shared with anyone but the co-researchers, and it was utilized solely for the registered research objectives.

## Results

From 3962 (weighted 3916.67) pregnant women, about 155 (15.21%) urban and 848 (29.29%) rural residences of the pregnant women did not use antenatal care services in 2019, whereas 602 (59.10%), and 1085 (37.47%) of urban and rural pregnant women used four and more antenatal care services, respectively. The mean and variance of observations among urban residents were 3.69 and 4.89, and they are 2.59 and 4.22 among rural residents (Figs 2 and 3) and (Table 2).

The sample mean of the ANC number of visits among urban and rural areas was 3.69 and 2.59, with a sample variance of 4.89 and 4.22, respectively. There is a greater variance than compared to the mean in both cases, which suggests an over-dispersion. Hence, the negative binomial regression model (NBRM) would be better for modeling the number of antenatal care visits among urban and rural settings (Table 2).

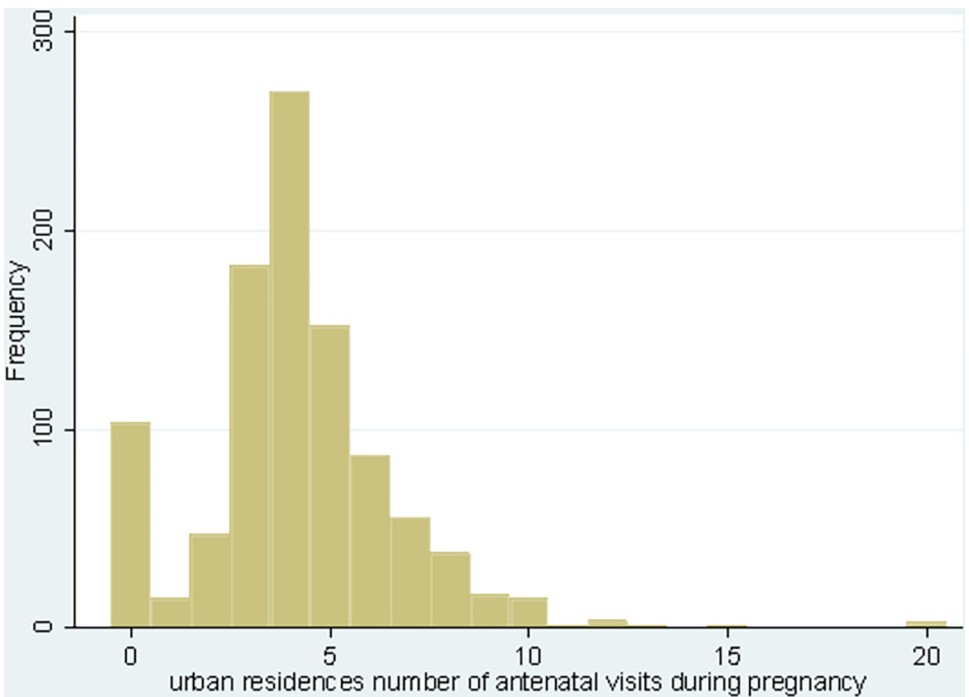

**Fig 2. A histogram showing the number of antenatal care visits among urban residents in Ethiopia, 2019.**

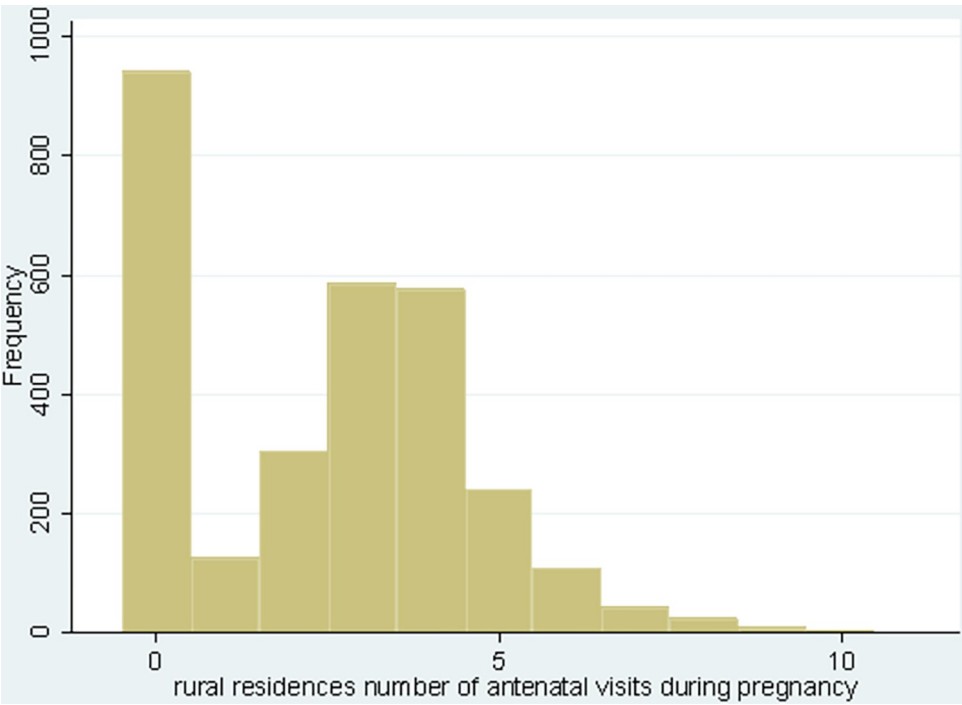

**Fig 3. A histogram showing the number of antenatal care visits among rural residents in Ethiopia, 2019.**

**Table 2. Frequency and percentage of ANC visits among urban and rural residents in Ethiopia in 2019.**

| Variables | Category | Urban | | Rural | |
|---|---|---|---|---|---|
| | | Weighted frequency | Percentage (%) | Weighted frequency | Percentage (%) |
| Number of ANC visits | No visits | 155.14 | 15.21 | 848.38 | 29.29 |
| | 1 | 19.86 | 1.95 | 110.38 | 3.81 |
| | 2 | 45.32 | 4.44 | 248.05 | 8.56 |
| | 3 | 196.85 | 19.30 | 604.31 | 20.86 |
| | 4 | 274.22 | 26.88 | 645.61 | 22.29 |
| | 5 | 155.13 | 15.21 | 249.80 | 8.62 |
| | 6 | 96.10 | 9.42 | 127.60 | 4.40 |
| | 7 | 40.30 | 3.95 | 25.52 | 0.88 |
| | 8 | 17.82 | 1.75 | 24.55 | 0.85 |
| | 9 | 10.97 | 1.08 | 6.12 | 0.21 |
| | 10+ | 8.28 | 0.81 | 6.35 | 0.22 |
| | Total | 1019.99 | 100 | 2,896.68 | 100 |
| Mean | | 3.69 | | 2.59 | |
| Variance | | 4.89 | | 4.22 | |
| Skewness | | 0.32 | | .22 | |
| Kurtosis | | 4.88 | | 2.55 | |
| Minimum | | 0 | | 0 | |
| Maximum | | 20 | | 11 | |
| Chi square | | $\chi 2 = 437.51, p < 0.001$ | | | |

## Magnitude of ANC service utilization between urban and rural pregnant women in Ethiopia, 2019

The frequency of ANC visits was higher for pregnant women in age groups 25–29 in both urban and rural areas than in other age groups. One-third 358 (36.09%) urban and 940 (31.65%) rural respondents had attained primary school. The utmost majority of respondents 442 (44.56%) in urban areas and 1,410 (47.47%) in rural areas were Muslim religion followers. Based on the urban and rural segregated wealth index, most respondents 385 (38.81%) in urban residence were the richest; whereas 987 (33.23%) of them were the poorest. In both urban 882 (88.91%) and rural 2,753 (92.69%) residences, married pregnant women hold the majority by current marital status (Table 3).

**Model selection technique for antenatal care utilization.** The estimates of the log likelihoods of both AIC and BIC slightly support the adoption of the NB model in both the urban and rural analyses. Of the two different models (Poisson and NB) being fitted, NB has the lowest AIC (4300.28) and BIC (4415.98) in both final models (Table 4).

## Factors associated with the number of antenatal care utilizations between urban and rural communities

By keeping other variables constant, in the Negative Binomial Poisson Model, maternal age, educational status, and household wealth index variables become significant predictors for the low frequency of ANC service utilization among urban residences. Similarly, educational status, religion, household wealth index, and marital status show a significant association with the frequency of antenatal care service utilization among rural residences.

As the age of the women increases by one year, the number of antenatal care utilization increases by 1.3, 1.56, 1.65, 1.66, 1.55, and 1.23 among 20–24 year-old women (IRR = 1.30,

**Table 3. Sociodemographic characteristics disparity of antenatal care service utilization between urban and rural communities in Ethiopia, 2019.**

| Variables | Category | Urban (N = 1019.99) | | | Rural (N = 2,896.68) | | | Test statistics |
|---|---|---|---|---|---|---|---|---|
| | | Weighted frequency | % | Mean(95%CI) | Weighted frequency | % | Mean(95%CI) | |
| **Age** | 15–19 | 52.21 | 5.12 | 2.81(1.35–4.28) | 174.90 | 6.04 | 2.18(1.72–2.64) | χ2 = 25.75 P = <0.01 |
| | 20–24 | 224.34 | 21.99 | 3.71(3.07–4.34) | 542.87 | 18.74 | 2.86(2.56–3.16) | |
| | 25–29 | 331.42 | 32.49 | 3.95(3.57–4.34) | 858.85 | 29.65 | 2.69(2.43–2.96) | |
| | 30–34 | 213.31 | 20.91 | 3.90(3.35–4.45) | 583.05 | 20.13 | 2.67(2.42–2.93) | |
| | 35–39 | 127.24 | 12.47 | 3.71(3.15–4.26) | 461.85 | 15.94 | 2.35(1.99–2.70) | |
| | 40–44 | 47.52 | 4.66 | 2.26(0.98–3.54) | 209.89 | 7.25 | 2.46(1.95–2.96) | |
| | 45–49 | 23.94 | 2.35 | 3.07(1.83–4.30) | 65.26 | 2.25 | 1.62(1.05–2.18) | |
| **Educational level** | No education | 303.82 | 29.79 | 2.67(2.20–3.14) | 1706.56 | 58.91 | 2.19(1.94–2.44) | χ2 = 596.14 P = <0.01 |
| | Primary | 420.33 | 41.21 | 3.77(3.27–4.27) | 990.57 | 34.20 | 3.01(2.81–3.21) | |
| | Secondary | 177.43 | 17.40 | 4.59(4.18–5.00) | 165.39 | 5.71 | 3.88(3.56–4.20) | |
| | Higher | 118.41 | 11.61 | 4.76(4.29–5.24) | 34.16 | 1.18 | 4.55(3.83–5.28) | |
| **Religion** | Orthodox | 392.96 | 38.53 | 4.38(4.08–4.69) | 1041.84 | 35.97 | 3.04(2.80–3.28) | χ2 = 28.18 P = <0.01 |
| | Protestant | 322.17 | 31.59 | 2.96(2.50–3.42) | 760.05 | 26.24 | 2.59(2.28–2.90) | |
| | Muslim | 300.61 | 29.47 | 3.57(3.20–3.95) | 1035.42 | 35.75 | 2.22(1.78–2.65) | |
| | Others[a] | 4.25 | 0.42 | 5.71(5.13–6.28) | 59.37 | 2.05 | 1.38(0.45–2.32) | |
| **Wealth index: urban/ rural** | Poorest | 191.09 | 18.73 | 2.16(1.75–2.56) | 623.87 | 21.54 | 1.51(1.27–1.76) | χ2 = 337.58 P = <0.01 |
| | Poorer | 167.76 | 16.45 | 2.64(1.83–3.44) | 589.46 | 20.35 | 2.37(2.02–2.72) | |
| | Middle | 225.72 | 22.13 | 3.53(3.16–3.90) | 586.09 | 20.23 | 2.75(2.40–3.09) | |
| | Richer | 219.61 | 21.53 | 4.63(4.29–4.97) | 545.98 | 18.85 | 2.94(2.68–3.20) | |
| | Richest | 215.81 | 21.16 | 5.12(4.70–5.54) | 551.27 | 19.03 | 3.55(3.32–3.79) | |
| **Marital status** | Never married | 5.39 | 0.53 | 5.10(3.11–7.09) | 15.45 | 0.53 | 1.34(0.27–2.40) | χ2 = 19.16 P = <0.01 |
| | Married | 921.72 | 90.37 | 3.77(3.43–4.10) | 2726.71 | 94.13 | 2.61(2.40–2.82) | |
| | Living with a partner | 5.96 | 0.58 | 3.74(3.25–4.23) | 21.38 | 0.74 | 2.68(1.35–4.02) | |
| | Widowed | 15.73 | 1.54 | 2.29(-0.79–5.37) | 27.86 | 0.96 | 2.11(1.18–3.04) | |
| | Divorced | 49.89 | 4.89 | 2.88(1.83–3.93) | 73.19 | 2.53 | 2.74(2.05–3.43) | |
| | No longer living together or separated | 21.29 | 2.09 | 3.37(2.25–4.49) | 32.10 | 1.11 | 1.93(1.18–2.68) | |

*(Continued)*

**Table 3.** (Continued)

| Variables | Category | Urban (N = 1019.99) | | | Rural (N = 2,896.68) | | | Test statistics |
|---|---|---|---|---|---|---|---|---|
| | | Weighted frequency | % | Mean(95%CI) | Weighted frequency | % | Mean(95%CI) | |
| **Number of living children** | 0 | 10.15 | 0.99 | 3.93(2.78–5.08) | 28.91 | 1.00 | 2.26(1.46–3.05) | χ2 = 192.43 P = <0.01 |
| | 1–2 | 538.96 | 52.84 | 4.13(3.81–4.45) | 1095.63 | 37.82 | 2.95(2.71–3.18) | |
| | 3–4 | 285.99 | 28.04 | 3.50(2.83–4.16) | 785.54 | 27.12 | 2.61(2.36–2.86) | |
| | 5 and above | 184.88 | 18.13 | 2.74(2.29–3.19) | 986.61 | 34.06 | 2.20(1.95–2.45) | |

*Catholic and traditional religion followers

95% CI: 1.05–1.61), 25–29 year-old women (IRR = 1.56, 95% CI: 1.27–1.92), 30–34 year-old women (IRR = 1.65, 95% CI: 1.33–2.05), and 35–39 year-old women (IRR = 1.55, 95% CI: 1.18–2.03), respectively, when compared with 15–19 years-old women among urban residences.

The utilization of ANC services increased with the educational level of the women in both urban and rural residences. The number of antenatal care visits increased with 1.18 (IRR = 1.18, 95% CI: 1.07–1.30), 1.26 (IRR = 1.26, 95% CI: 1.13–1.42), and 1.25 (IRR = 1.25, 95% CI: 1.11–1.41) times higher as the educational level increased with one unit among primary, secondary, and higher educated women than no education women in urban residences, respectively. Whereas, as the educational level increases with one unit, antenatal care visits increases by 1.34 (IRR = 1.34, 95% CI: 1.24–1.45), 1.54 (IRR = 1.54, 95% CI: 1.34–1.76), and 1.58 (IRR = 1.58, 95% CI: 1.28–1.95) times higher among primary, secondary, and higher educated women than no education women in rural residences, respectively.

A number of ANC visits were also associated with respondents' religion. So that as a woman becomes a Protestant, Muslim, and Others (Catholic and Traditional) religion follower respondent, the ANC service utilization decreases by 24% (IRR = 0.76, 95% CI: 0.69–0.83), 21% (IRR = 0.79, 95% CI: 0.73–0.85), and 44% (IRR = 0.56, 95% CI: 0.43–0.71) less as compared to the Orthodox religion follower respondents among urban residences.

Concerning wealth index, for the middle, richer, and richest respondents, the number of ANC visits increased by 1.31 (IRR = 1.31, 95% CI: 1.13–1.52), 1.45 (IRR = 1.45, 95% CI: 1.26–1.66), and 1.68 (IRR = 1.68, 95% CI: 1.46–1.93) times more, respectively, as the wealth index increased by one unit than the poorest respondent among urban residences. While, as the wealth index increased by one unit, the frequency of antenatal care visits among pregnant women residing in poorer households increased by 1.51 (IRR = 1.51, 95% CI: 1.37–1.67), middle 1.66 (IRR = 1.66, 95% CI: 1.50–1.83), richer 1.71 (IRR = 1.71, 95% CI: 1.55–1.91), and richest 1.89 (IRR = 1.89, 95% CI: 1.72–2.09) times than the poorest respondent in rural residences.

**Table 4.** Test statistics of the model fit.

| Model | Urban | | | | Rural | | | |
|---|---|---|---|---|---|---|---|---|
| | AIC | BIC | Log-likelihood (df) | LR test (p-value) | AIC | BIC | Log-likelihood (df) | LR test (p-value) |
| **PR** | 4300.97 | 4408.76 | -2128.48 | 359.99(0.00) | 12111.93 | 12243.85 | -6033.97 | 1006.69(0.00) |
| **NBR** | 4300.28 | 4412.98 | -2127.14 | 293.87(0.00) | 11793.16 | 11931.08 | -5873.58 | 498.02(0.00) |

AIC: Akaike's Information Criterion; BIC: Bayesian Information Criterion; NBR: Negative Binomial Regression; PR: Poisson Regression

**Table 5. Factors associated with the number of ANC service utilizations between urban and rural communities in Ethiopia, 2019.**

| Variables | Category | Urban | Rural |
|---|---|---|---|
| | | IRR(95% CI) | IRR(95% CI) |
| **Age** | 15–19 | 1 | 1 |
| | 20–24 | 1.30(1.05–1.61)* | 1.03(0.89–1.19) |
| | 25–29 | 1.56(1.27–1.92)* | 1.09(0.94–1.26) |
| | 30–34 | 1.65(1.33–2.05)* | 1.15(0.98–1.35) |
| | 35–39 | 1.66(1.33–2.09)* | 1.1(0.93–1.32) |
| | 40–44 | 1.55(1.18–2.03)* | 1.05(0.87–1.29) |
| | 45–49 | 1.23(0.83–1.81) | 0.92(0.70–1.21) |
| **Educational level** | No education | 1 | 1 |
| | Primary | 1.18(1.07–1.30)* | 1.34(1.24–1.45)* |
| | Secondary | 1.26(1.13–1.42)* | 1.54(1.34–1.76)* |
| | Higher | 1.25(1.11–1.41)* | 1.58(1.28–1.95)* |
| **Religion** | Orthodox | 1 | 1 |
| | Protestant | 0.92(0.84–1.02) | 0.76(0.69–0.83)* |
| | Muslim | 0.93(0.86–1.01) | 0.79(0.73–0.85)* |
| | Other | 1.10(0.65–1.85) | 0.56(0.43–0.71)* |
| **Wealth index: urban/rural** | Poorest | 1 | 1 |
| | Poorer | 1.16(0.98–1.39) | 1.51(1.37–1.67)* |
| | Middle | 1.31(1.13–1.52)* | 1.66(1.50–1.83)* |
| | Richer | 1.45(1.26–1.66)* | 1.71(1.55–1.91)* |
| | Richest | 1.68(1.46–1.93)* | 1.89(1.72–2.09)* |
| **Marital status** | Never married | 1 | 1 |
| | Married | 0.94(0.67–1.32) | 1.85(1.19–2.86)* |
| | Widowed/divorced | 0.83(0.58–1.18) | 1.95(1.24–3.07)* |
| **Number of living children** | 0 | 1 | 1 |
| | 1–2 | 1.24(0.92–1.67) | 1.34(0.99–1.80) |
| | 3–4 | 1.11(0.82–1.51) | 1.27(0.93–1.73) |
| | 5 and above | 1.05(0.76–1.45) | 1.28(0.94–1.75) |

1: reference category

*: significant variable at α: 5%, IRR: Incidence rate ratio, Urban LogL: -2127.14, Rural LogL: -5873.58

As women become currently married and widowed or separated, their ANC service utilization gets higher by 1.85 (IRR = 1.85, 95% CI: 1.19–2.86) and 1.95 (IRR = 1.95, 95% CI: 1.24–3.07), respectively, than the respondent who was never married among rural residences (Table 5).

## Discussion

The study revealed that the mean antenatal care visits and utilization of pregnant women in urban residences were higher than in rural residences. This finding is in line with the study by Enyew and Mekonnen [31], who reported that women living in urban areas had a higher expected number of antenatal care visits during their pregnancy than women living in rural areas. Our result also aligns with the existing evidence in Angola [10]. In this study, there was a discrepancy in ANC utilization among urban and rural pregnant women in Ethiopia. This was consistent with the findings of the Vietnam study, which stated that antenatal care utilization was not similar in urban and rural areas [9]. The difference in remoteness, road and

transport access, accessibility of healthcare facilities, skilled medical staff, and quality of service in healthcare facilities could be plausible explanations [32]. In addition, other than the health service-related factors, pregnant women and their spouses' socioeconomic, educational status, media exposure, and health-seeking behavior could affect ANC utilization in urban areas more than in rural areas.

According to this finding, there was a significant association between ANC utilization and age group. Age groups of pregnant women 20–24, 25–29, 30–34, 35–39, 40–45 were more likely to utilize ANC than the 15–19 age group of pregnant women in urban residences. This was consistent with the previous study in Ethiopia [31], which found older pregnant mothers (in the age range of 35 to 49 years) in 2011 used ANC more frequently. It is also in line with the study conducted in Tanzania [33] and Rwanda [34], which showed that pregnant women between the ages of 15–19 years were more likely to use antenatal care services than those over the age of 19. Furthermore, this finding was inconsistent with a study conducted in Nepal [35], which revealed that women over the age of 35 were less likely to seek prenatal care, and a prior study [22], which showed no connection between antenatal care service use in Ethiopia and younger age.

In this study, pregnant women's educational level was an important factor that determined ANC utilization regardless of the type of residence. Previous studies [14, 31, 36, 37] also revealed that the number of ANC visits is related to the educational level of pregnant women. According to this study, pregnant women with primary, secondary, and higher education were more likely to use antenatal care services in both urban and rural residences in Ethiopia. According to a number of studies [10, 38, 39], women who have completed elementary school or higher are more likely to feel confident acting on their own health issues and are more aware of the benefits of using health services, including ANC utilization. What is more, this finding was in agreement with a study conducted in Nepal, which discovered a strong and significant relationship between education for women and increased use of ANC services [35].

We observed that the frequency of antenatal care utilization significantly varies according to their religious status among pregnant women in urban Ethiopia. The current finding was consistent with prior available studies [40–42], which found that the use of antenatal care services varies depending on religion. This may indicate that some religious institutions, like the Orthodox Church, may have been encouraging their followers to use healthcare services to attain good health.

We found that the women's wealth index was significantly associated with the use of ANC services in both urban and rural residences in Ethiopia. Wealthier pregnant women tend to use ANC services more frequently. The higher ANC utilization among wealthier or richest pregnant women in Ethiopia may be because the poorest women cannot afford the non-medical costs like transportation related to using antenatal care services [43], even though the medical costs for ANC services were free there. Socio-economic issues may make it difficult for the poorest pregnant mothers to access ANC services, which could reduce the number of ANC visits by increasing the length of time that ANC is used [10]. Additionally, various studies [44–46] have demonstrated how socioeconomic status influences the use of ANC services in developing countries.

Another important factor identified by this study was that ANC utilization is highly associated with the current marital status of pregnant women in rural residences. More ANC visits were made by married and cohabiting pregnant women than by women who had never been in a relationship. This finding is in agreement with the study conducted in Rwanda [47]. This demonstrated that men's contributions to antenatal care visits and their wives' encouragement to use healthcare services were significant. Available previous studies also mentioned that married women or women in union may get partner support to attend ANC [48, 49].

The strength of the current study includes the use of negative binomial Poisson analysis analyses to overcome the over-dispersion nature of EMDHS data and deploying nationally weighted representative and most recent EMDHS data, which shows the country-level count of ANC service utilization and its associated factors among women of childbearing age. It also tries to identify the independent factors in urban and rural settings, which most of the studies didn't look into. The study also has some limitations. Showing a temporal relationship between ANC service utilization and its predictors was impossible due to the type of study design, cross-sectional, used for the survey. Additionally, because it was a small report, the EMDHS data did not provide details regarding several determinants of ANC service utilization. However, the researchers believe that the aforementioned limitations cannot significantly impair the validity of the study's conclusions.

## Conclusion

This study found significant disparities in antenatal care utilization among pregnant women in urban and rural areas. Rural pregnant women attended ANC relatively later, made fewer visits, and used ANC services considerably less frequently than their urban counterparts. It is recommended that the national strategy be updated and put into action with more specific guidelines and evaluation indicators to improve rural women's use of ANC services through the safety-net lens.

## Supporting information

**S1 File. STROBE-checklist.**
(DOCX)

**S2 File. DHS datasets authorization letter.**
(PDF)

## Acknowledgments

We acknowledge Dr. Senahara Korsa Wake for his direction, helpful offers, and support, and the Demographic Health Survey program office for allowing us to access all the relevant DHS data for this study.

## Author Contributions

**Conceptualization:** Fitsum Endale, Belay Negassa, Senahara Korsa Wake.

**Data curation:** Fitsum Endale, Belay Negassa, Tizita Teshome, Addisu Shewaye, Beyadiglign Mengesha, Endale Liben.

**Formal analysis:** Fitsum Endale, Belay Negassa, Addisu Shewaye, Endale Liben, Senahara Korsa Wake.

**Methodology:** Fitsum Endale, Tizita Teshome, Addisu Shewaye, Beyadiglign Mengesha, Endale Liben, Senahara Korsa Wake.

**Resources:** Tizita Teshome.

**Software:** Fitsum Endale, Beyadiglign Mengesha, Senahara Korsa Wake.

**Visualization:** Fitsum Endale, Beyadiglign Mengesha.

**Writing – original draft:** Fitsum Endale, Belay Negassa.

**Writing – review & editing:** Fitsum Endale, Belay Negassa, Tizita Teshome, Addisu Shewaye, Beyadiglign Mengesha, Endale Liben, Senahara Korsa Wake.

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
