## [Decision Letter · Decision Letter 0]

14 Nov 2023

PONE-D-23-11412Antenatal care service utilization disparities between urban and rural communities in Ethiopia: a negative binomial Poisson regression of 2019 Ethiopian Demography Health SurveyPLOS ONE

Dear Dr. Endale,

Thank you for submitting your manuscript to PLOS ONE. After careful consideration, we feel that it has merit but does not fully meet PLOS ONE’s publication criteria as it currently stands. Therefore, we invite you to submit a revised version of the manuscript that addresses the points raised during the review process.

We look forward to receiving your revised manuscript.

Kind regards,

Obasanjo Afolabi Bolarinwa, Masters

Academic Editor

PLOS ONE

Journal Requirements:

Reviewers' comments:

Reviewer's Responses to Questions

**Comments to the Author**

1. Is the manuscript technically sound, and do the data support the conclusions?

Reviewer #1: Yes

Reviewer #2: No

2. Has the statistical analysis been performed appropriately and rigorously? 

Reviewer #1: Yes

Reviewer #2: No

3. Have the authors made all data underlying the findings in their manuscript fully available?

Reviewer #1: Yes

Reviewer #2: Yes

4. Is the manuscript presented in an intelligible fashion and written in standard English?

Reviewer #1: Yes

Reviewer #2: No

5. Review Comments to the Author

Reviewer #1: The article on variation of ANC in urban and rural Ethiopia is interesting or order to make policy informed decision on reduction of mother and child death. Please see below observations

1. line 60-61 show the progress made in reducing MMR globally

2. Make the narrative sequential. After highlighting the global issues on ANC, discuss the issue as if affect developing countries, then Africa before focusing in Ethiopia (Line 67)

3. Line 72 Fourth trimester is basically not for monitoring fetal condition, except in cases of pregnancy elongation.

4. Move line 73-76 to paragraph 2

5. Line 89 Please state the previous studies refer

6. Line 94 state or show the previous study

7. Line 95-98 where was these studies conducted

8. line 98 Ten years ago (Kindly specify year e.g 2023)

9. Line 103-105 Please recast as this is not the first research on the subject. Highlight the gap that this study fill

10. Line 255-256 is not clear

11. Line 255-264 state the reason for high ANC in urban region

12. Line 286-288 state tat ANC varies by religious affiliation. Do not state the type of religion

Reviewer #2: Abstract

The abstract extends far beyond the word limit required by the journal which should be about 300 words and I feel that the authors need to abide by this word limit.

The background section of the abstract does not seem to present any rationale for this study which should inform the objective of the study. This is because obviously, there are numerous studies in the context of Ethiopia that have explored ANC utilization and geographical disparities inherent in it. Providing the justification for the study here in the background will give the readers a quick idea of why the study is important and needed.

The result section presented in the abstract will need to be written again to present the most important and significant result, which will ensure that the authors align with the 300-word limit required by the journal.

The conclusion section of the abstract seems like a summary of the results already presented in the previous section. The authors should provide a better conclusion for their study and provide policy or programmatic recommendations or implications for future studies in Ethiopia.

Introduction

The paper will require professional editing services to rid it of various grammatical issues.

Lines 73-76 can be added to the previous paragraph.

The information in lines 83-88 should be added to lines 67-72 because the information being discussed in both sections is similar and falls under the same theme of ANC recommendations and utilization.

Most earlier studies talked about in lines 77-78, what have they found in urban areas? Is the aim of this study to compare evidence from rural areas with these studies??

Has there been no study in Ethiopia where the level of ANC in rural and urban areas has been compared? I am afraid that you have not provided enough justification for your study and this needs to be clearly provided in this study. Also, the gap could be methodological, if previous studies have not employed count models to study ANC, you can fill that gap in addition to what you have discussed as the justification for the present study.

Methodology

To avoid going beyond the journal word limit, I think there isn’t any need to provide much information on the study setting. Rather, those words can be used to provide more information about the EDHS data (check the 2019 EDHS report for more information).

On what basis were the study variables (independent variables) selected for this study? Did the authors consider the need to include variables are the household and community level considering that the study aims to compare urban and rural contexts?

The authors need to provide more information on the outcome variable. How was the question asked in the survey? What did the response from the respondents look like? Is the response a count variable??

While it is understandable that your data is over-dispersed and violates the assumptions of Poisson regression, hence the need for other count models. I think it will be good to better inform your readers about the reason behind your decision to choose negative binomial regression. What does the distribution of your outcome variable look like? Is the outcome variable zero-inflated? These and more are issues to discuss in order for NB regression to be considered the most suitable.

Also, the authors should provide more information on model selection criteria.

The authors can also provide a bar chart showing the distribution of the outcome variable [ANC].

It would also be great to see the authors present mathematical notations of their NB regression models (see previous studies that have employed NB regression for this).

Similar studies that have employed count models to study ANC that could provide guidance to the authors

https://link.springer.com/article/10.1007/s40745-021-00328-x

https://www.tandfonline.com/doi/full/10.1080/03630242.2016.1222325

https://journals.sagepub.com/doi/full/10.1177/24551333211030349

https://journals.plos.org/plosone/article?id=10.1371/journal.pone.0228215

Results

The authors should consult some other studies that have employed count models for the best practices in the interpretation of the results of count models, especially the IRR.

6. PLOS authors have the option to publish the peer review history of their article (what does this mean?). If published, this will include your full peer review and any attached files.

Reviewer #1: **Yes: **Chukwudeh Stephen Okechukwu

Reviewer #2: **Yes: **Oluwatobi A. Alawode

---

## [Author Response · Author response to Decision Letter 0]

5 Feb 2024

AUTHORS’ RESPONSES TO EDITORS AND REVIEWERS

Dear Obasanjo Afolabi Bolarinwa (Academic Editor of PLOS ONE),

We really thank you for your thorough reading and constructive comments and suggestions on our manuscript and for the opportunity to revise and resubmit it for the second time. We are pleased to submit the revised research article “Antenatal care service utilization disparities between urban and rural communities in Ethiopia: a negative binomial Poisson regression of 2019 Ethiopian Demography Health Survey” for your consideration on PLOS ONE. On the following page, you will find our response to the editor’s and reviewers comments. On behalf of my co-authors, I want to thank you again for your consideration of this resubmission. We appreciate your time and look forward to your response.

Sincerely, 

Fitsum Endale (BSc, MSc) (corresponding author)

fitsumale@gmail.com

Authors’ Response to Editor

PONE-D-23-11412

Antenatal care service utilization disparities between urban and rural communities in Ethiopia: a negative binomial Poisson regression of 2019 Ethiopian Demography Health Survey

PLOS ONE

Dear Dr. Endale,

Thank you for submitting your manuscript to PLOS ONE. After careful consideration, we feel that it has merit but does not fully meet PLOS ONE’s publication criteria as it currently stands. Therefore, we invite you to submit a revised version of the manuscript that addresses the points raised during the review process.

If applicable, we recommend that you deposit your laboratory protocols in protocols.io to enhance the reproducibility of your results. Protocols.io assigns your protocol its own identifier (DOI) so that it can be cited independently in the future. For instructions see: https://journals.plos.org/plosone/s/submission-guidelines#loc-laboratory-protocols. Additionally, PLOS ONE offers an option for publishing peer-reviewed Lab Protocol articles, which describe protocols hosted on protocols.io. Read more information on sharing protocols

at https://plos.org/protocols?utm_medium=editorial-email&utm_source=authorletters&utm_campaign=protocols.

We look forward to receiving your revised manuscript.

Kind regards,

Obasanjo Afolabi Bolarinwa, Masters

Academic Editor

PLOS ONE

Journal Requirements:

(https://doi.org/10.1371/journal.pone.0230416)? If you’ve not already done so, consider depositing your raw data in a repository to ensure your work is read, appreciated and cited by the largest possible audience. You’ll also earn an Accessible Data icon on your published paper if you deposit your data in any participating repository (https://plos.org/open-science/open-data/#accessible-data).

Response: Thank you again for taking the time to manage the publication process and reminding us to follow the journal's submission guidelines; we have attempted to submit the revised version of the manuscript as per your advice, direction, and suggestions.

END__________________________________

THANK YOU!!!

AUTHORS’ RESPONSE TO REVIEWERS

Dear Reviewers,

We thank you for a thorough reading and for your constructive comments and suggestions on our manuscript, as well as for the opportunity to revise and resubmit. We are pleased to submit the revised manuscript “Antenatal care service utilization disparities between urban and rural communities in Ethiopia: a negative binomial Poisson regression of 2019 Ethiopian Demography Health Survey” for your consideration in PLOS ONE. On the following page, you will find our response to the reviewers’ comment. On behalf of my co-authors, I thank you for your consideration of this resubmission again. We appreciate your time and constructive comments once again.

Sincerely, 

Fitsum Endale (BSc, MSc) (corresponding author)

fitsumale@gmail.com

AUTHORS’ RESPONSE TO REVIEWER #1

Comment to the Author: The article on variation of ANC in urban and rural Ethiopia is interesting or order to make policy informed decision on reduction of mother and child death. Please see below observations

1. Line 60-61 show the progress made in reducing MMR globally

Response: As per the given comment, we have included a statement that shows the progress in reducing MMR up to 2020.

2. Make the narrative sequential. After highlighting the global issues on ANC, discuss the issue as if affect developing countries, then Africa before focusing in Ethiopia (Line 67)

Response: We have rewrote the introduction in a manner that keeps the sequence from global to local.

3. Line 72 Fourth trimester is basically not for monitoring fetal condition, except in cases of pregnancy elongation.

Response: As per the comment, we have corrected the indicated sentence.

4. Move line 73-76 to paragraph 2

Response: We have rearranged the paragraph as per the given comment.

5. Line 89 Please state the previous studies refer

Response: we have changed the way of reporting and the paragraph as a whole

6. Line 94 state or show the previous study

Response: We have tried to rewrite the paragraph in a manner that shows the overall contradiction.

7. Line 95-98 where was these studies conducted

Response: We have tried to indicate the place where the whole study was conducted. 

8. Line 98 Ten years ago (Kindly specify year e.g. 2023)

Response: We have corrected it as per the given comment.

9. Line 103-105 Please recast as this is not the first research on the subject. Highlight the gap that this study fill

Response: We have rewrite the justification as per the given direction.

10. Line 255-256 is not clear

Response: We have rewrote the sentence in a manner that makes it clear.

11. Line 255-264 state the reason for high ANC in urban region

Response: We have highlighted a reason or factors that may have caused the prevalence to be high.

12. Line 286-288 state that ANC varies by religious affiliation. Do not state the type of religion

Response: We have omitted the different religious groups’ names from the sentence and rewrote it as a general.

Response: Thank you very much for taking the time to review our work and for your positive feedback. We received your thoughtful review, along with helpful feedback and suggestions, as a valuable contribution to our work.

END___________________________________

THANK YOU!!!

AUTHORS’ RESPONSE TO REVIEWERS #2

Comments to the Author: 

Abstract

The abstract extends far beyond the word limit required by the journal which should be about 300 words and I feel that the authors need to abide by this word limit.

Response: We have concisely written the abstract part of the manuscript to be in the word limit of the journal.

The background section of the abstract does not seem to present any rationale for this study which should inform the objective of the study. This is because obviously, there are numerous studies in the context of Ethiopia that have explored ANC utilization and geographical disparities inherent in it. Providing the justification for the study here in the background will give the readers a quick idea of why the study is important and needed.

Response: We have also tried to highlight the gap in the abstract introduction as per the given comment.

The result section presented in the abstract will need to be written again to present the most important and significant result, which will ensure that the authors align with the 300-word limit required by the journal.

Response: We have amended the result part of the abstract in a short and precise way to indicate the pertinent findings.

The conclusion section of the abstract seems like a summary of the results already presented in the previous section. The authors should provide a better conclusion for their study and provide policy or programmatic recommendations or implications for future studies in Ethiopia.

Response: We also tried to include a policy implication of the study in the conclusion part of the abstract and the main section as well.

Introduction

The paper will require professional editing services to rid it of various grammatical issues.

Response: In terms of language proficiency, we contacted language experts from our university's language and literacy department to check it and try to address the issues.

Lines 73-76 can be added to the previous paragraph.

Response: The indicated paragraph was merged and rewritten as per the comment.

The information in lines 83-88 should be added to lines 67-72 because the information being discussed in both sections is similar and falls under the same theme of ANC recommendations and utilization.

Response: paragraphs with similar ideas were merged as per the given direction.

Most earlier studies talked about in lines 77-78, what have they found in urban areas? Is the aim of this study to compare evidence from rural areas with these studies??

Response: The previous studies, disparities in ANC utilization among rural and urban settings were indicated, and the factors associated were also listed. 

Has there been no study in Ethiopia where the level of ANC in rural and urban areas has been compared? I am afraid that you have not provided enough justification for your study and this needs to be clearly provided in this study. Also, the gap could be methodological, if previous studies have not employed count models to study ANC, you can fill that gap in addition to what you have discussed as the justification for the present study.

Response: We have rewrote the justification of the study as per the given comment by indicating the methodological gap and providing current evidence that compares the two population groups.

Methodology

To avoid going beyond the journal word limit, I think there isn’t any need to provide much information on the study setting. Rather, those words can be used to provide more information about the EDHS data (check the 2019 EDHS report for more information).

Response: We have reduced the extra explanation and included the statement as given in the comment.

On what basis were the study variables (independent variables) selected for this study? Did the authors consider the need to include variables are the household and community level considering that the study aims to compare urban and rural contexts?

Response: We have tried to indicate the way the study included the independent variables as factors-those which contribute to the factors of ANC utilization based on previous studies and available in the already collected data.

The authors need to provide more information on the outcome variable. How was the question asked in the survey? What did the response from the respondents look like? Is the response a count variable??

Response: We have indicated the outcome variable is a discrete response and have tried to describe how the response variable was measured and how the respondents responded.

While it is understandable that your data is over-dispersed and violates the assumptions of Poisson regression, hence the need for other count models. I think it will be good to better inform your readers about the reason behind your decision to choose negative binomial regression. What does the distribution of your outcome variable look like? Is the outcome variable zero-inflated? These and more are issues to discuss in order for NB regression to be considered the most suitable. Also, the authors should provide more information on model selection criteria.

Response: A fascinating and significant issue has been brought forward. Just to be reasonable, we have indicated how the model was selected, and since we have been focusing on the application of the classical poison models, we were limited to the Poisson and negative binomial models only.

The authors can also provide a bar chart showing the distribution of the outcome variable [ANC]. It would also be great to see the authors present mathematical notations of their NB regression models (see previous studies that have employed NB regression for this).

Similar studies that have employed count models to study ANC that could provide guidance to the authors

https://link.springer.com/article/10.1007/s40745-021-00328-x

https://www.tandfonline.com/doi/full/10.1080/03630242.2016.1222325

https://journals.sagepub.com/doi/full/10.1177/24551333211030349

https://journals.plos.org/plosone/article?id=10.1371/journal.pone.0228215

Response: After reviewing the studies suggested by the reviewers, we have included negative binomial regression mathematical model notations.

Results

The authors should consult some other studies that have employed count models for the best practices in the interpretation of the results of count models, especially the IRR.

Response: We have rewritten the IRR interpretation of the results as per the given direction. 

Response: Thank you very much for all of your thoughtful comments and recommendations, as well as your kind words, which have helped us refine our work in its final form for the readers.

END___________________________________

THANK YOU!!!

---

## [Decision Letter · Decision Letter 1]

26 Feb 2024

Antenatal care service utilization disparities between urban and rural communities in Ethiopia: a negative binomial Poisson regression of 2019 Ethiopian Demography Health Survey

PONE-D-23-11412R1

Dear Dr. Endale,

We’re pleased to inform you that your manuscript has been judged scientifically suitable for publication and will be formally accepted for publication once it meets all outstanding technical requirements.

Kind regards,

Obasanjo Afolabi Bolarinwa, Masters

Academic Editor

PLOS ONE

Additional Editor Comments (optional):

Reviewers' comments:

Reviewer's Responses to Questions

**Comments to the Author**

1. If the authors have adequately addressed your comments raised in a previous round of review and you feel that this manuscript is now acceptable for publication, you may indicate that here to bypass the “Comments to the Author” section, enter your conflict of interest statement in the “Confidential to Editor” section, and submit your "Accept" recommendation.

Reviewer #1: All comments have been addressed

Reviewer #2: All comments have been addressed

2. Is the manuscript technically sound, and do the data support the conclusions?

Reviewer #1: Yes

Reviewer #2: Yes

3. Has the statistical analysis been performed appropriately and rigorously? 

Reviewer #1: I Don't Know

Reviewer #2: Yes

4. Have the authors made all data underlying the findings in their manuscript fully available?

Reviewer #1: Yes

Reviewer #2: No

5. Is the manuscript presented in an intelligible fashion and written in standard English?

Reviewer #1: Yes

Reviewer #2: Yes

6. Review Comments to the Author

Reviewer #1: Thank you for updating the manuscript. Well Done. Kindly proofread the manuscript for English Language and context to ensure that it aligns with global standards.

Reviewer #2: The authors have attended to the issues identified in the article during the first round of review. Paper can be read again for grammatical and stylistic issues.

7. PLOS authors have the option to publish the peer review history of their article (what does this mean?). If published, this will include your full peer review and any attached files.

Reviewer #1: **Yes: **Chukwudeh Okechukwu Stephen

Reviewer #2: No

---

## [Editor Report · Acceptance letter]

5 Mar 2024

PONE-D-23-11412R1 

PLOS ONE

Dear Dr. Endale, 

I'm pleased to inform you that your manuscript has been deemed suitable for publication in PLOS ONE. Congratulations! Your manuscript is now being handed over to our production team.

Kind regards, 

on behalf of

Dr. Obasanjo Afolabi Bolarinwa 

Academic Editor

PLOS ONE